# Using paleoclimate reconstructions to analyse hydrological epochs associated with Pacific Decadal Variability

LANYING ZHANG, GEORGE KUCZERA

*School of Engineering, University of Newcastle, Callaghan, New South Wales, Australia*

ANTHONY S. KIEM

*Centre for Water, Climate and Land (CWCL), Faculty of Science, University of Newcastle, Callaghan, New South Wales, Australia*

GARRY WILLGOOSE

*School of Engineering, University of Newcastle, Callaghan, New South Wales, Australia*

## Abstract

The duration of dry or wet hydrological epochs (run lengths) associated with positive or negative Inter-decadal Pacific Oscillation (IPO) or Pacific Decadal Oscillation (PDO) phases, termed Pacific Decadal Variability (PDV), is an essential statistical property for understanding, assessing and managing hydroclimatic risk. Numerous IPO and PDO paleoclimate reconstructions provide a valuable opportunity to study the statistical signatures of PDV, including run lengths. However, disparities exist between these reconstructions making it problematic to determine which reconstruction(s) to use to investigate pre-instrumental PDV and run length. Variability and persistence on centennial scales are also present in some millennium long reconstructions, making consistent run length extraction difficult. Thus, a robust method to extract meaningful and consistent

run lengths from multiple reconstructions is required. In this study, a dynamic threshold framework to
account for centennial trends in PDV reconstructions is proposed. The dynamic threshold framework
is shown to extract meaningful run length information from multiple reconstructions. Two
hydrologically important aspects of the statistical signatures associated with the PDV are explored: (i)
whether persistence (i.e. run lengths) during positive epochs is different to persistence during negative
epochs and (ii) whether the reconstructed run lengths are stationary during the past millennium.
Results suggest that there is no significant difference between run lengths in positive and negative
phases of PDV and that it is more likely than not that the PDV run length has been non-stationary in
the past millennium. This raises concerns about whether variability seen in the instrumental record
(the last ~100 years), or even in the shorter 300-400 year paleoclimate reconstructions, is
representative of the full range of variability.
Key words: PDV; run length; multiple reconstructions; stationarity

# 1. Introduction

A pattern of Pacific ocean-atmosphere climate variability at decadal time scales has been
identified and explored at various locations around the world including Africa (Reason and Rouault,
2002;Hoell and Funk, 2014), Eastern and Southern Asia (Krishnan and Sugi, 2003;Ma, 2007),
America (Mantua and Hare, 2002;Andreoli and Kayano, 2005) and Australia (Kiem et al.,
2003;Verdon et al., 2004;Verdon and Franks, 2006;Henley et al., 2011). This pattern, referred to in
this paper as Pacific Decadal Variability (PDV), is associated with sea surface temperature
fluctuations and sea level pressure changes in the north and south Pacific Ocean.
PDV is usually described by the Inter-decadal Pacific Oscillation (IPO) or Pacific Decadal
Oscillation (PDO) indices (Mantua et al., 1997;Power et al., 1999).  Of particular hydrological
relevance are the statistical characteristics of PDV phases where a phase refers to a period during
which the PDV index lies above (or below) some thresholds (Dong and Dai, 2015;Verdon et al.,
2004;Henley et al., 2011;Vance et al., 2015). The duration of a PDV phase (termed run length) is
defined as the time between consecutive crossings of the threshold. The phase may be described as
positive (negative) if the PDV index is above (below) the threshold, or dry (wet) if the PDV phases
are associated with predominantly dry (wet) hydrological conditions.

4        Of particular relevance to the hydrology and water resources community is the evidence that

the PDV phases can be associated with multi-decadal periods of persistently wetter or drier conditions
and corresponding increases in flood or drought risk in affected regions, particularly those on the
Pacific rim. PDV has been found to be related to a number of hydrological variables including
precipitation, streamflow, flood/drought risk (Cook et al., 2013;Kiem et al., 2003;Verdon et al.,
2004;Dai, 2013;Goodrich and Walker, 2011;Hu and Huang, 2009;Li et al., 2012;McCabe et al.,
2012;Mehta et al., 2011;Wang et al., 2014;Henley et al., 2013). For example, Kiem and Franks (2004)
showed in a case study for eastern Australia that the probability of reservoir storages falling below a
critical level differs significantly depending on the PDV phases (see Figure 6 in Kiem and Franks
(2004)). Kiem and Franks (2003) and Micevski et al. (2006) demonstrated that flood risk in eastern
Australia is strongly dependent on PDV phase. Henley et al. (2013) extended this work to show that
short-term drought risk is strongly dependent not only on the PDV phase but also on the time spent in
a particular phase.

17        Despite the clear relevance of PDV phases to hydrological risk assessment and water resource

management, there remain considerable knowledge gaps about the statistical characteristics of PDV,
including run lengths. The instrumental record shows that PDV phases have varied irregularly with
runs ranging from less than a decade to several decades during the past century. However, the
instrumental record is insufficient to characterize the statistical characteristics of PDV run lengths. In
response to this, significant advances have been made in reconstructing pre-instrumental PDV
behaviour to extend data length. For example, at least 12 IPO or PDO reconstructions have been
published (Biondi et al., 2001;Gedalof and Smith, 2001;D'Arrigo et al., 2001;MacDonald and Case,
2005;D'Arrigo and Wilson, 2006;Shen et al., 2006;Verdon and Franks, 2006;Linsley et al.,
2008;Mann et al., 2009;McGregor et al., 2010;Henley et al., 2011;Vance et al., 2015).
In view of these published PDV reconstructions, the fundamental question is whether useful
information can be extracted about the statistical characteristics of PDV persistence. In dissecting this
question, several unresolved issues are identified:
1)    Static threshold methods have been used to estimate run lengths of PDV phases. This
raises the concern that biased conclusions about the statistical characteristics of decadal climate
variability may be drawn when reconstructions exhibit centennial or longer trends. The non-
parametric Mann-Whitney test method was used in Verdon and Franks (2006) whereby a crossing
was defined when the test detected a statistically significant difference between two halves of data in
a 30-year moving window, with zero used as a static threshold to define the sign of PDV phases
(Verdon-Kidd, personal communication, 24 April, 2017). Henley et al. (2011) used the instrumental
mean of the composite IPO/PDO index. After standardizing the indices, Vance et al. (2015) used ±0.5
as thresholds to determine crossings and the sign of PDV phases, and Henley et al. (2017) used the
long-term modelled  IPO Tripole Index (TPI) mean as the static threshold with run lengths less than 5
years omitted. However, when static threshold methods are used, extraordinary long centennial run
lengths may be identified in the reconstructions that exhibit centennial or longer trends (MacDonald
and Case, 2005;Mann et al., 2009). Analysis based on data with long-term trends can lead to results
that are overwhelmed by such trends and hence efforts have been made to detrend data without losing
useful signals (Wahl et al., 2006;von Storch et al., 2006;Wu et al., 2007). As stated by von Storch et al.
(2006), "it is commonly accepted that proxy indicators may contain nonclimatic trends".  Moreover,
the calibration and validation of any statistical method using nondetrended data may be compromised
because the nonclimatic trends may be interpreted as a climate signal. The centennial trends in PDV
reconstructions may be either nonclimatic trends or non-decadal climate trends. Whichever the case, it
is necessary to filter such centennial trends before interpreting decadal climate variability. Given that
static threshold methods cannot remove trends, how should the run length extraction method be
designed to extract useful and consistent information from all reconstructions, including
reconstructions exhibiting centennial trends? To what extent are run length distributions sensitive to
the choice of extraction parameters (e.g. threshold, window width etc.)?

2)	The durations of positive and negative PDV phases have not been treated separately in most previous research (e.g. Verdon and Franks, 2006; Linsley et al., 2008; Henley et al., 2011a ) Using a high-resolution millennial-length IPO reconstruction, Vance et al. (2015) observed that positive phases dominate and last longer than negative phases. Owing to nonlinear hydrologic feedback mechanisms, such as elasticity of streamflow to runoff (Chiew, 2006) and shifted rainfall-runoff relationships (Saft et al., 2015), the duration of increased (decreased) precipitation during a wet (or dry) PDV phase may impact on streamflow in a highly nonlinear manner. Therefore, assuming wet and dry runs follow the same distribution may misrepresent the intensity of impacts. Is there sufficient evidence about the dissymmetry between positive and negative phase run lengths from the multiple reconstructions? Will run length samples from each phase lead to significantly different run length simulations?

3)	Stationarity of climate signatures is a necessary, yet implicit, assumption underlying many analyses based on paleoclimate reconstructions – and hydroclimate stochastic modelling used in water resource assessment (e.g. Henley et al., 2013). However, a number of studies have suggested the possibility of climatic non-stationarity over the past millennium. Phipps et al. (2013) found that the relationships between paleoclimate proxies and climatic variables have been changing at least the last millennium. Razavi et al. (2015) explored the stationarity of the mean, variance and autocorrelation structures of paleo tree-ring proxy data and concluded that the key statistical characteristics of climate had undergone significant change over time. The existence of extraordinary warm and cold periods, known as Medieval Climate Anomaly (~950-1250CE) and Little Ice Age (~1400-1700CE) (Mann et al., 2009;Phipps et al., 2013;Atwood et al., 2015), also challenges the assumption of climatic stationarity. Furthermore, externally forced anthropogenic climate change, arising from elevated atmospheric concentrations of greenhouse gases, may also make recent and future climate different from the long past (Kirtman et al., 2013). Therefore, care should be taken when positing that PDV characteristics are stationary. Statistical signatures of past reconstructed PDV runs should be tested for non-stationarity (and where possible non-stationarity should be attributed to causal mechanisms) to ensure robust and representative estimation of the full range of variability.

In view of the above mentioned issues, this study has the following objectives:

2         1)      To develop a more robust run length extraction method that is applicable to all

reconstructions, including those with centennial trends. Of particular importance is the sensitivity of
the inferred run length distributions to the choice of the extraction method parameters.

5         2)      To identify hydrologically important statistical properties of PDV that are common to

different reconstructions. In particular two fundamental questions are explored: (i) whether
persistence (i.e. run lengths) during positive epochs is different to persistence during negative epochs
and (ii) whether the reconstructed run lengths are stationary during the past millennium. Both
questions address fundamental concerns about using PDV and run length information in stochastic
hydroclimatic models used to assess drought and flood risk.

11        The rest of the paper is organized as follows. Section 2 briefly introduces the instrumental and

reconstructed PDV records used in this research. The dynamic threshold run length extraction method
is introduced in Section 3, along with its comparison with the static threshold method and
determination of its parameters. The hydrologically important statistical signatures of PDV are
analysed and discussed in Section 4, with conclusions presented in Section 5.

# 2. Data

## 2.1 Instrumental PDV records

18        A number of instrumental IPO and PDO indices have been published to characterize PDV

since ~1900. There is a strong correlation between IPO and PDO indices, suggesting that both indices
represent a similar broad pattern of climate variability. The primary difference between the IPO and
PDO indices is that the IPO index is based on a broader spatial scale than the PDO index (Folland et
al., 2002;Verdon and Franks, 2006;Henley et al., 2011).

1        Three PDV indices are used in this study: (1) The unfiltered instrumental PDO index

2        (http://research.jisao.washington.edu/pdo/PDO.latest.txt), denoted as PDO_Mantua, is the monthly

3        standardized value of the leading principal component of monthly Sea Surface Temperature (SST)

4        anomalies in the North Pacific Ocean, poleward of 20N (Mantua et al., 1997;Zhang et al., 1997); (2)

5        The unfiltered monthly IPO index from Parker et al. (2007), denoted as IPO_Parker, is the second

6        covariance empirical orthogonal function of low-pass-filtered SST; (3) The unfiltered monthly Tripole

7        Index (TPI) for the IPO (Henley et al., 2015), denoted as IPO_Henley, is based on the SST anomalies

8        in three large geographic regions of the Pacific. Annual (calendar year) values of these three indices

9        are taken as the average of monthly values.

10       Figure 1 shows time series of the three indices along with probability density plot, quantile-

11       quantile (QQ) plot and autocorrelation plots. The distribution density plot and QQ plot of the annual

12       PDV indices suggest that the instrumental PDV indices are approximately normally distributed with

13       zero mean and unit standard deviation (except for IPO_Parker). The lag-one autocorrelation lies

14       between about 0.3 and 0.5. By lag three, the autocorrelations lie within the 95% confidence bands for

15       zero autocorrelation.

16       The instrumental PDV indices are used for two purposes in this study:

17       (a) to assess the similarity between IPO and PDO indices. If instrumental IPO and PDO

18       indices can be used to represent the same broad pattern of climate variability - and noting that

19       different reconstructions are calibrated against different instrumental indices - there is no need to

20       recalibrate all reconstructions against the same instrumental data.

21       (b) to guide the selection of the parameters of the run length extraction method. If selected

22       parameters can identify credible run lengths in the instrumental PDV record, it is assumed that they

23       can also produce credible run lengths in the reconstructed record.

## 2.2 PDV reconstructions

Twelve published annual PDV reconstructions from different sites with different proxies are used in this study. Of these, nine reconstructed the last 300-400 years and three reconstructed the last ~1000 years. The reconstructions are based on paleo proxies (i.e. preserved physical characteristics of the environment that can be directly measured) from the northern and southern, and western and eastern regions of the Pacific Ocean. A summary of these reconstructions is presented in Table 1, while Figure 7 in Section 3.2.3 presents time series plots of the 12 reconstructed indices, and they are denoted throughout by the four-character author name and two-character publishing year. It is worth noting that Mann09 was not directly calibrated to an instrumental PDV index, and also has a negative correlation with other reconstructions. However, because the correlation of this reconstruction with the other ones is relatively high (Henley, 2017), the Mann09 reconstruction is retained with the sign reversed.

# 3. Run length extraction

## 3.1 Static and dynamic threshold run length extraction methods

However, for reconstructions Macd05, Mann09 and Vanc15 which exhibit centennial trends (see Table 1 and Figure 2), the above mentioned static threshold methods used in Verdon and Franks (2006), Henley et al. (2011), Vance et al. (2015) and Henley et al. (2017) may not identify meaningful decadal phases. Extraordinary long run lengths are identified if the overall mean is used as a threshold with some runs lasting several centuries (see Figure 2). It is not clear whether these centennial trends are due to low frequency climate variability or due to unknown local factors affecting the proxy. With such reconstructions, one can either exclude them from analysis or filter out the centennial trends.

Exclusion forgoes any useful information from the reconstructions. On the other hand, filtering out
the centennial trends offers the prospect of extracting possibly useful information about run lengths.

3        We adopt the latter approach proposing a dynamic threshold framework to filter out

centennial trends. In this framework, potential crossings are taken to be the change points detected by
the Mann-Whitney test, and the sign of PDV phases is determined by a dynamic threshold that takes
centennial trends into consideration - this relaxes the restriction of a static threshold defining a phase
crossing. Figure 2 provides insight about the mechanics of dynamic threshold method showing the
PDV time series and the resulting block phase waveform. The key steps of this framework are:

9        1.    Detect step-change points. For a given reconstruction, apply a change point detection

method. A number of methods can be used. In this study we used the non-parametric Mann-Whitney
test method (Mauget, 2003) with a given window width w and confidence level α to identify the step-
change points. The method involves centring a window of width w at a particular year t and then
applying the Mann-Whitney test to the samples of width w/2 at or before and after year t. A step
change is deemed to occur if the Mann-Whitney test statistic lies outside the α confidence limits
(under the null hypothesis).

16        2.    Merge consecutive step-change points. When two or more step change points occur in

consecutive years, replace them with a single change point. That is, if there are either *2n-1* or *2n* years
that are determined as consecutive change points (*n*=1, 2, …), replace them with one change point at
year *n*. This guarantees that the new change point is always closest to the middle of the run of
consecutive step-change points.
3.    Assign a phase to each run defined by the interval bounded by two step-change points. Let $i(t)$

22        denote the PDV index in year $t$, $i_f(t)$ the filtered PDV index using a first-order Butterworth filter

23        (which was used by Henley et al. (2011) to filter paleo PDV indices) with cut-off frequency $1/y$

24        year$^{-1}$, $t_{cp}^k$ the year that the $k^{th}$ change point occurs (from steps 1 and 2) and $s(t)$ the phase state of

25        year $t$. The mean PDV index for the $k^{th}$ run is

$$\bar{i}^{\,k} = \frac{1}{(t_{cp}^{k} - t_{cp}^{k-1} + 1)} \sum_{t_{cp}^{k-1}}^{t_{cp}^{k}} i(t) \tag{1}$$

while the corresponding mean of the filtered index is

$$\bar{i}_{f}^{\,k} = \frac{1}{(t_{cp}^{k} - t_{cp}^{k-1} + 1)} \sum_{t_{cp}^{k-1}}^{t_{cp}^{k}} i_{f}(t) \tag{2}$$

The phase of the $k^{th}$ run length is then defined by

$$s(t) = \begin{cases} 1 \text{ if } \bar{i}^{\,k} \geq \bar{i}_{f}^{\,k} \\ 0 \text{ if } \bar{i}^{\,k} < \bar{i}_{f}^{\,k} \end{cases}, t \in (t_{cp}^{k-1}, t_{cp}^{k} - 1) \tag{3}$$

4.  Determine run lengths using the time series $s(t)$. The run length of the $k^{th}$ run will be
$t_{cp}^{k} - t_{cp}^{k-1}$ .
This dynamic threshold method can be seen as a generalization of previous run length
extraction methods. If parameter $y$ is set to the total number of years in a given reconstruction, the
dynamic threshold method reduces to the method used in Verdon and Franks (2006). If change points
are defined by the PDV index crossing the threshold and the cut-off frequency parameter $y$ of the first-
order Butterworth filter is set to the total number of years in the reconstruction, the dynamic threshold
reduces to the method used in Henley et al. (2011) and Henley et al. (2017).
Three parameters need to be specified in the dynamic threshold method: window width $w$ and
confidence level α in the Mann-Whitney test, and cut-off frequency $y$ of the Butterworth filter. The
standard cut-off frequency parameter in the dynamic threshold method is used to filter out centennial
trends that may be mixed with decadal variability and should have little influence on the statistical
characteristics of the inferred runs. A cut-off frequency of 1/100 years is considered to adequately
meet these requirements. Hence, $y=100$ is used in this study. In this study, we set the confidence level
to be 90%, a value that is consistent with other reconstruction studies (eg Gedalof and Smith, 2001;

Shen et al., 2006; McGregor et al., 2010; Pent et al., 2015). It has been shown that reconstructions are sensitive to the choice of the window used in the Mann-Whitney test as well as the choice of threshold (Henley et al., 2017;Henley, 2013). Accordingly, this study investigates the sensitivity of dynamic threshold method results to the choice of window width $w$ in the Mann-Whitney test. This sensitivity analysis is used to guide the selection of parameters to be used in the analysis of all 12 PDV reconstructions.

## 3.2 Results from different run length extraction methods

### 3.2.1 Comparison of static and dynamic threshold

To further demonstrate the shortcomings of the static threshold method and the need for a dynamic threshold method, run length samples are extracted using the dynamic threshold method proposed in this study (denoted as "Dynamic" in the figures) and the static threshold method used in Verdon and Franks (2006) (denoted as "Static" in the figures). These two methods only differ in step 3 of Section 3.1. In the static method, the overall mean defined as $\overline{i_f} = \frac{1}{n}\sum_{1}^{n} i_f(t)$ (in which $n$ is number of years in corresponding reconstruction) is used as the threshold instead of the dynamic threshold defined in equation (2). Extracted run length samples using the dynamic and static threshold methods from 3 millennium long reconstructions with centennial trends are plotted in Figure 2. Black lines are filtered standardized PDV indices; green lines represent the Butterworth (1/100) filtered data in the dynamic plots and the overall mean in the static plots; red lines represent the run extracted using either the dynamic or static thresholds method. The run length distributions are plotted in Figure 3.

It is shown in Figure 2 that extraordinarily long runs are identified when the static threshold method is used - the stronger the centennial trends, the longer the runs. On the other hand, use of the

dynamic threshold framework appears to filter centennial trends and produce more meaningful run
lengths. The run length density plots show that the dynamic threshold method for the Macd05 and
Mann09 reconstructions removes centennial-scale runs. In the case of Vanc15 the centennial trends
are muted, possibly because Vanc15 is annually resolved and very accurately dated (Vance et al.,
2015) and as such is more likely to exhibit more realistic annual to decadal variability than Macd05
and Mann09 (see also Figure 2). Individual reconstructions are subject to different error structures and
magnitudes. Use of multiple reconstructions can help reduce the impact of errors, and thus provide
more insight into the statistical characteristics of PDV run lengths. Nonetheless, errors still need to be
carefully considered. This is illustrated by Henley et al. (2011) who combines individual
reconstructions according to their accuracy. However, the accuracies of the individual reconstructions
are not high (Nash-Sutcliffe indices <0.48) suggesting that even favouring the more accurate
individual reconstructions will still result in a combined/overall reconstruction with large errors.
Therefore, the focus of our research is to identify a signal (or signals) that is (are) common to all
individual reconstructions.
**3.2.2 Window width sensitivity analysis**
Henley (2013) demonstrated that the mean run length is strongly dependent on the choice of
window width in Mann-Whitney method and concluded that a subjective choice of window width is
likely to bias the resulting run length distributions. To explore the sensitivity of window width on run
length distributions, run length boxplots for all reconstructions are plotted in Figure 4 for window
widths from 10 to 60 years with step of 2, with positive and negative samples plotted together and
separately.
Figure 4 shows that the run length distributions clearly depend on the choice of window width
with longer window widths leading to longer run length samples. However, the change is gradual
particularly for shorter window widths. As a result, run length distributions are not particularly
sensitive to small variations in window width of the order of 10 years.

### 3.2.3 Determination of parameters and application of dynamic threshold method

To guide the selection of an operationally useful window width, several window widths are applied to the three instrumental PDV time series to identify which produces results that match existing knowledge about run length in instrumental periods. Four window widths, 10, 20, 30 and 40 years, are applied to the instrumental time series with the extracted runs plotted in Figure 5. Interpretation of Figure 5 is illustrated for Figure 5d. The green line represents the IPO_Henley index, from which three runs, denoted by green circles, are extracted: ~1900-1938 (positive), ~1938-1976 (negative), ~1976-2004 (positive). These are very similar (in number, sign and duration) to the black (PDO_Mantua) and red run lengths (IPO_Parker). It can be seen that a window width of 10 years identifies very short runs which are more likely to represent random fluctuations rather than different PDV phases  - we note that Henley et al. (2017) discarded runs with lengths less than 5 years. However, for the longer window widths of 20, 30 and 40 years, the differences become more muted.

To better understand these differences, consider window widths of 20 and 40 years. When the 40-year window is used, the first 20 years of the window are compared to the last 20 years. As a result, runs of less than 20 years may be overlooked. To demonstrate this, different window widths are applied to all reconstructions to extract run lengths, from which conditional run length distributions are determined. Figure 6 presents run length distributions in six panels. Panels (a1) to (a3) compare distributions for 20 and 40-year windows. Panel (a1) shows the empirical density of run lengths, while panels (a2) and (a3) show the empirical densities conditioned on run lengths less than 20 years and greater than 20 years respectively. Likewise Panels (b1) to (b3) repeat this sequence for 20 and 60-year windows with the conditioning threshold of 30 years.

Panels (a1) and (b1) show that the unconditional run length distributions are different using different window widths with the difference greater for a greater difference in window width. However, panels (a3) and (b3) show that the conditional distributions for the longer runs are very similar. This shows that both the short and long window widths extract the longer runs in a largely

consistent manner. However, the choice of window width strongly affects the distribution of shorter
runs as shown in panels (a2) and (b2). Overall, this demonstrates that shorter window widths can "see"
higher frequency runs better than longer window widths but both "see" the same distribution of lower
frequency runs.

5       These considerations suggest that the window width should be as small as possible, yet

sufficiently big to filter out runs that are considered more likely to be random fluctuations rather than
PDV phases. Therefore, in this study, a 20-year window is used to ensure decadal or longer runs are
identified. Figure 7 plots the extracted runs for the 12 reconstructions using the dynamic threshold
method with 90% confidence limits and 20-year window in the Mann-Whitney test. The PDV indices
plotted in Figure 7 were filtered using a Butterworth filter with a cut-off frequency of $1/11$ years$^{-1}$ to
more clearly present the decadal variability in each reconstruction. In the case of Verd06, no PDV
indices are available, so the original runs are plotted. Visual inspection of Figure 7 suggests that the
identified runs in all reconstructions seem largely consistent with the multi-decadal fluctuations in the
filtered PDV indices. In the case of reconstructions with pronounced centennial trends (such as
Macd05 and Mann09), positive and negative runs tend to match multi-decadal fluctuations in the
filtered indices even when the indices persist above or below the long-term average over centennial
scales.

# 4. Statistical signatures of PDV run lengths

19       The dynamic threshold method allows run lengths to be extracted from all the PDV

reconstructions, even from those with centennial trends. If we had a perfect reconstruction there
would be no need for any further reconstructions. However, the fact is that each paleoclimate
reconstruction is subject to errors, both random and systematic, that are not fully understood.
Therefore, it is pertinent to identify statistical features that are common to the reconstructions and also
important to hydroclimatic risk assessments. This addresses the second objective of this study. In
particular, the analysis seeks to answer the following questions: Are the distributions of positive and
negative PDV run lengths statistically different, and is the variability seen in the instrumental record
(the last ~100 years), or the shorter 300-400 year paleoclimate reconstructions, representative of the
full range of variability that has occurred in the past or of what is plausible in the future? These
questions are of particular importance for predicting near-term PDV phase behaviour and assessing
near-term flood and drought risk and therefore are the primary focus of this section.

## 4.1 Are run lengths different during positive and negative PDV phases?

8         Vance et al. (2015) analysed an IPO reconstruction from AD 1000 to 2003 and concluded that

the positive IPO phase (IPO>0.5) has an average duration of 14 years and the negative IPO phase
(IPO< −0.5) has an average duration of 9 years. This is the first known analysis that considered
positive and negative PDV phases separately.

12         To explore the hypothesis that run lengths have different distributions during positive and

negative PDV phases, boxplots of run lengths for each reconstruction are shown in Figure 8. Panel (a)
shows boxplots for all run lengths for each reconstruction, Panel (b) shows boxplots of pooled
positive and negative run lengths, while Panel (c) shows boxplots of positive and negative run lengths
for each reconstruction. Panel (b) suggests that the pooled median positive and negative run lengths
are similar, but that positive run lengths are more variable. However, Panel (c) shows that there is
considerable variability between reconstructions. The absence of a consistent difference between
positive and negative run distributions may be a consequence of sampling variability masking any
underlying difference. Sampling variability refers to the variability in the target statistics that arises
when random sampling is repeated. Therefore, it is important to recognize that differences in run
statistics may be due to sampling variability. It is only when differences are bigger than what would
be expected due to sampling variability that we would conclude there is a significant difference. This
seems reasonable given that the average number of phases (either positive or negative) per
reconstruction is only 13. A more formal statistical analysis is required to explicitly deal with
sampling variability arising from the small samples.

3        Henley et al. (2011) adopted the Gamma distribution as the probability model of PDV run

lengths. The Gamma distribution has two parameters which are related to the mean $\mu$ and standard
deviation $\sigma$ of the run lengths. For each reconstruction, the positive and negative runs are extracted
and the posterior distribution of the mean and standard deviation is inferred using MCMC (Gelman
and Rubin, 1992). Figure 9 presents the joint posterior distribution of the differences in
mean($\mu^+ - \mu^-$ _ and standard deviation ($\sigma^+ - \sigma^-$) for each reconstruction. If the difference between
positive and negative runs is statistically significant, one would expect the posterior distribution to be
well removed from the zero-difference origin.

11       Inspection of Figure 9 suggests there is no strong and consistent evidence to reject the

assumption that positive and negative phase run length distributions are the same. Although the mean
of positive phase run lengths tends to be longer than negative phase run lengths in several
reconstructions (e.g. Vanc15, Henl11, Lins08, Darr01), the statistical significance of this difference is
weak and the phenomenon is not consistent across all reconstructions. A similar conclusion applies to
the standard deviation of positive and negative phase run lengths. This highlights the fundamental
limitation of the small PDV run samples derived from the reconstructions.

## 18   4.2 Are run lengths stationary over the last millennium?

19       With paleoclimate reconstructions becoming longer, questions arise whether the PDV run

length has been stationary in the past millennium and whether a shorter paleoclimate reconstruction is
representative of the full range of variability that has occurred in the past or of what is plausible in the
future. The stationarity issue has been explored most PDV reconstructions used in this study. Biondi
et al. (2001) identified weakened amplitude of bi-decadal oscillations in the late 1700s to mid-1800s.
Based on reconstruction over 1700-1979, D'Arrigo et al. (2001)  declared that variations at around 12-
17 years are considerably more pronounced from 1700-1849 relative to 1850-1979 with a shift

towards decreased amplitude since about 1850. Using visual inspection, Gedalof and Smith (2001) stated that the 30-70 year PDV frequency is confined to the pre-1840 portion of the series over the period of 1599-1983. D'Arrigo and Wilson (2006) reported a broad range of lower (multi-decadal to centennial) frequencies over the 1565-1988 period. Shen et al. (2006) pointed out that the major PDO regime timescale modes of oscillation have not been persistent over 1470-1998 and that 75-115 year and 50 –70 year oscillations dominated the periods before and after 1850, respectively.  Linsley et al. (2008) argued that decadal to inter-decadal variability in the south Pacific convergence zone region has been relatively constant over 1650-2004. MacDonald and Case (2005) presented evidence for a strong and persistent negative PDO state during the medieval period (AD 900 to 1300), suggesting a cool north eastern Pacific at that time. Mann et al. (2009) observed that the Little Ice Age (LIA, 1400 to 1700) and the Medieval Climate Anomaly (MCA, 950 to 1250) showed extra variability and persistence, challenging the assumption of a stationary climate in the past millennium. The mechanism responsible for the extra variability and persistence in studies done by MacDonald and Case (2005) and Mann et al. (2009) is unclear and may be explained by centennial climate variability. It is unknown whether the PDV structure in the past millennium has remained stationary during periods that display extra variability and persistence. This issue can be investigated by filtering out the centennial trends.

All the above stationarity studies are based on single reconstructions. A further complication arises for reconstructions that only cover the last ~300-400 years where the underlying non-stationarity may be masked as a consequence of sampling variability or the sampling variability is misinterpreted as non-stationarity. To mitigate this, here we use only millennium-length records, Macd05 (993-1996), Mann09 (500-2006) and Vanc15 (1000-2003), in an attempt to obtain statistically meaningful conclusions on PDV stationarity. The millennium-length records are split into pre-1600 and post-1600 samples to explore whether run length characteristics are statistically similar in these two periods and to shed light on whether shorter records (either the ~100 year instrumental record or the 300-400 year paleoclimate reconstructions) are able to represent the full range of

variability. The year 1600 is selected as the change point because most of the shorter PDV

reconstructions date back to ~1600 (refer to Table 1 and Figure 2 for details).

Figure 10 presents boxplots of pre-1600 and post-1600 run lengths: panel (a) presents

boxplots of pooled pre-1600 and post-1600 run lengths, while panel (b) shows boxplots of pre-1600

and post-1600 runs for each of the three millennium-length reconstructions. A consistent pattern

emerges in which the median run length and interquartile range are greater in the pre-1600 period for

both pooled samples (Figure 10a) and each individual reconstruction (Figure 10b).

Bearing in mind that the samples are small, a Gamma distribution was inferred for the pre-

and post-1600 samples. Figure 11 presents the joint posterior distribution of the differences in the pre-

and post-1600 means and standard deviations for each reconstruction using all run lengths. A

consistent pattern once again emerges. For all three reconstructions the posteriors of the pre- and post-

1600 differences lie in the lower left quadrant with the origin minimally intersecting with the posterior.

Therefore, the evidence that there is a difference is visually strong.

## 4.3 Discussion

All of the reconstructions reported in Table 1 did not distinguish between positive and

negative PDV runs in their analysis, except for Vance et al. (2015). Based on a millennium IPO

reconstruction, Vance et al. (2015) found that IPO has an average positive phase (IPO>0.5) duration

of 14 years and a negative phase (IPO< −0.5) duration of 9 years over A.D.1000-2003, and concluded

that positive and negative phases durations and frequencies were different in the last millennium. This

is the first study that addresses positive and negative PDV runs separately. However, based on

comparisons with other reconstructions our results show that there remains uncertainty as to whether

or not the run lengths of positive epochs are statistically different to the run length of negative epochs.

It is important to keep in mind that the analysis here is fundamentally limited by small PDV run

length sample sizes from most of the reconstructions. Two theories may explain the absence of a

consistent difference between positive and negative run distributions. One is that no statistically

meaningful differences exist and the other is that differences do exist but are masked by sampling
variability. The latter seems plausible given the average number of phases (either positive or negative)
per reconstruction are only 13.
All three millennium-length records lead to higher inferred run length mean and standard
deviation in the pre-1600 samples, suggesting longer and more varied PDV runs during the period
before 1600 AD. The fact that the differences in the mean and standard deviation of run lengths pre-
and post-1600 appear to be statistically significant raises an important question, should the
information from pre-1600 reconstructions be used to infer PDV behaviour in the near climate. If one
adopts a gradualist view of climate non-stationarity, the immediate past would be considered more
representative of the present than the more distant past. Setting aside for the moment the possibility
that the climate-proxy relationship may be not be stationary, the gradualist perspective would support
discarding the information from the pre-1600 reconstruction on the grounds that it introduces bias
when inferring a statistical model of near climate PDV runs. We offer two reasons opposing this
perspective.
First, there is no assurance that climate non-stationarity evolves in a gradual manner so that
PDV behaviour in the near climate is better represented by the post-1600 climate than the pre-1600
climate. Indeed there is evidence to the contrary, namely that non-stationarity may be characterized by
seemingly shorter-term shifts. Ho et al. (2017) found that American streamflow in the twentieth
century featured longer wet and dry spells compared to the preceding 450 years, suggesting the
possibility of occurrence of extended dry or wet periods in the future that exceeds variability
presented in instrumental and short (i.e. < 500 years) paleoclimate reconstructions. Similar
conclusions are also drawn in other studies (Vance et al., 2015;Tozer et al., 2016;Ho et al., 2015b, a).
Second, the inclusion of pre-1600 records offers a significantly larger sample of run lengths
and therefore is more likely to capture the full range of what is plausible. Extended wet/dry periods
considered as rare and extreme (or even impossible) based on recent (i.e. instrumental) history might
actually be more likely than thought (or even common) when put into context of climate conditions
seen over the last 1000-2000 years. Even if one accepts the gradualist view of non-stationarity, the
increased information about PDV more than outweighs the potential bias arising from the use of an
apparently statistically inconsistent record. At least until such time as it is proven that the climate has
totally shifted and that what occurred in the past is no longer possible – also required is identification
of when that shift occurred.

6        It is therefore clear that in addition to short instrumental records inadequately sampling the

length and severity of dry/wet epochs, the shorter paleoclimate reconstructions may similarly
misrepresent hydroclimatic variability and persistence. For instance, from Ho et al. (2015a), when
using the period 1684–1980, both the driest and wettest instrumental decadal rainfall lies closer to the
middle of the minimum and maximum decadal reconstructed rainfall range. However, using the same
method, Ho et al. (2015a) found that when 2751 years of reconstructions were examined, the wettest
instrumental decadal rainfall is near the bottom of the maximum decadal reconstructed rainfall range,
while the driest instrumental decadal rainfall is near the top of the minimum decadal reconstructed
rainfall range. This suggests that statistics inferred from either instrumental data or short paleoclimate
reconstructions will underestimate the variability contained in the longer palaeoclimate
reconstructions. Therefore, research focused on developing and analysing information about
hydroclimatic conditions over the last 1000 years or more (e.g. multi-millennium paleoclimate
reconstructions) is critical if we are to better understand, quantify and manage the full range of
hydroclimatic variability, and associated risks, that are possible in the future.

20       This study is affected by two fundamental limitations. One is the small PDV run sample size

from the reconstructions. For instance, the average number of phases (either positive or negative) per
reconstruction are only 13. Another limitation is that all statistical signatures are based on
paleoclimate reconstructions. Although multiple reconstructions are used, this study is constrained by
the intrinsic limitations of paleoclimate reconstruction based interpretations, such as assumptions of
stationarity of the proxy-climate relationship (Phipps et al., 2013). The possibility that all these

reconstructions are biased in the same direction cannot be ruled out and as such conclusions based on

multiple reconstructions may also be biased.

# 5. Conclusions

PDV, and associated run lengths of predominantly dry or wet conditions, has profound

implications for precipitation/streamflow prediction, flood/drought risk assessment and water resource

management. Therefore, a better understanding of the statistical characteristics of PDV is needed.

However, because instrumental records are short (~100 years at best in Australia), there is

considerable uncertainty about the key statistical signatures of PDV, including run lengths.

Paleoclimate reconstructions, serving as a vehicle to provide longer realizations of PDV, have been

widely studied and used. However, for various reasons (e.g. proxy sources, site locations, proxy

resolutions, reconstruction methods and local non-PDV effects) temporal coherence between different

reconstructions varies significantly (Kipfmueller et al., 2012). Hence, one PDV reconstruction may

lead to significantly different conclusions from another reconstruction.

The aim of this paper was to explore the characteristics of key statistical signatures of PDV

based on multiple PDV reconstructions. We focused on the duration of dry or wet hydrological

epochs (i.e. run lengths) as the key statistical signature and developed a robust method for extracting

run lengths from multiple reconstructions. Extracting run lengths objectively is challenging, given

interactions with sources of variability on a variety of temporal and spatial scales. For instance, when

millennium-length reconstructions are used, variability and persistence at the centennial scale can lead

to biased characterisation of decadal climate variability. The dynamic threshold framework introduced

in this study, which takes centennial trends into consideration, was shown to extract meaningful run

length information from multiple reconstructions.

No strong evidence was found to support the assumption that run lengths have statistically

different distributions during positive and negative PDV phases. Analysis based on three millennium

long reconstructions suggests that it is more likely than not that PDV run length has been non-

stationarity in the past millennium. This again highlights that the instrumental record (~100 years at
best), and even short paleoclimate reconstructions (i.e. less than 400 years into the past), should not be
assumed to represent the full range of variability that has occurred in the past or what may occur in
the future. Caution should be exercised regarding assumptions that the climate is stationary and the
implications of a non-stationary climate should at least be tested. Longer climate reconstructions (i.e.
1000 years or more) appear to give more useful information and insights into what has occurred in the
past and also tell us more about what is plausible and what needs to be planned for in the future.
All of the reconstructions explored in this study have focused on the run length of PDV, yet
they have provided quite different run length characterizations. This again highlights that each
reconstruction is subject to errors, both random and systematic, that are not fully understood.
Therefore, a challenging and important research direction is the analysis of run length characteristics
with explicit consideration of the errors. Another important research direction is the continued
refinement of methods that utilize PDV information in water resource management as we work
toward improving how we assess and manage hydrological risks in a variable and changing climate.

# 15 Acknowledgments

Funding for this research was provided by Australian Research Council Linkage Grant
LP120200494 with further funding and/or in-kind support also provided by the NSW Office of
Environment and Heritage, Sydney Catchment Authority, Hunter Water Corporation, NSW Office of
Water, and NSW Department of Finance and Services.

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

# Tables

**Table 1 Available PDV paleoclimate reconstructions and their descriptions**

| Reconstruction | Abbreviation | Period | Location | Proxy |
|---|---|---|---|---|
| Biondi et al. (2001) | Bion01 | 1661-1991 | North Pacific (North America) | Tree ring based PDO |
| D'Arrigo et al. (2001) | Darr01 | 1700-1979 | Northeast Pacific (Western North America) | Tree ring based PDO |
| Gedalof and Smith (2001) | Geda01 | 1599-1983 | North Pacific (Coastal western North America) | Tree ring based PDO |
| MacDonald and Case (2005) | Macd05 | 993-1996 | North Pacific (South California and Canada) | Tree ring based PDO |
| D'Arrigo and Wilson (2006) | Darr06 | 1565-1988 | North Pacific (Eastern Asian) | Tree ring based PDO |
| Shen et al. (2006) | Shen06 | 1470-1998 | Western North Pacific (Eastern China) | Proxy data of Summer Rainfall |
| Verdon and Franks (2006) | Verd06 | 1662-1998 | Both North and South Pacific | Other reconstructions based PDO |
| Linsley et al. (2008) | Lins08 | 1650-2004 | South Pacific (Fiji and Tonga) | Oxygen isotope from coral cores based IPO |
| Mann et al. (2009) | Mann09 | 500-2006 | Global | Global tree ring, ice core, coral, sediment, and other assorted proxy records |
| McGregor et al. (2010) | Magr10 | 1650-1977 | Global | low frequency variability of the proxies of ENSO |
| Henley et al. (2011) | Henl11 | 1471-2000 (filtered) | Both North and South Pacific | Other reconstructions |
| (Vance et al. (2015)) | Vanc15 | 1000-2003 (filtered) | East Antarctica | Law Dome ice core based IPO |

# A list of figure captions

Figure 1 Annual instrumental time series, probability density plot, QQ plot and autocorrelation plot: (a): PDO from Mantua (1900-2015); (b): IPO from Parker (1871-2007); (c): IPO from Henley (1870-2007)

Figure 2 Run length time series extracted using the dynamic and static threshold methods for reconstructions with centennial trends

Figure 3 Comparison of run length distributions extracted using the dynamic and static threshold methods in reconstructions with centennial trends

Figure 4 Boxplot of run lengths samples from all reconstructions with different window width in Mann-Whitney method

Figure 5 Time series plots of different instrumental PDV indices and corresponding run lengths (represented by dots in corresponding colour) extracted using different window: (a) 10; (b) 20; (c) 30; (d) 40 years

Figure 6 Density plot of run length samples by applying different window widths to all reconstructions, (a1-a3): comparison between window width 20 and 40, in which (a1) is unconditional run length distribution, panel (a2) is the distribution conditioned on run lengths less than 20 years and panel (a3) is the distribution conditioned on run lengths greater than 20 years; (b1-b3): comparison between window width 20 and 60, in which (b1) is unconditional run length distribution, panel (b2) is the distribution conditioned on run lengths less than 30 years and panel (b3) is the distribution conditioned on run lengths greater than 30 years

Figure 7 Filtered PDV reconstruction time series (black line) with extracted run length (red line)

Figure 8 Boxplot of run lengths samples from different reconstructions: (a) all run lengths for each reconstruction; (b) pooled positive (grey) and negative (white) run lengths; (c) positive (grey) and negative (white) run lengths for each reconstruction

Figure 9 Density and scatter plots of the differences between mean and standard deviation of positive and negative run length simulations (positive minus negative)

Figure 10 Boxplot of run lengths samples from different reconstructions (white box indicates pre-1600 and grey box indicates post-1600): (a) pooled run lengths from all three reconstructions; (b) run lengths for each reconstruction

Figure 11 Density and scatter plots of the differences between mean and standard deviation of pre-1600 and post-1600 run length simulations (post-1600 minus pre-1600): (a) Macd05; (b) Mann09; (c) Vanc15

# Figures

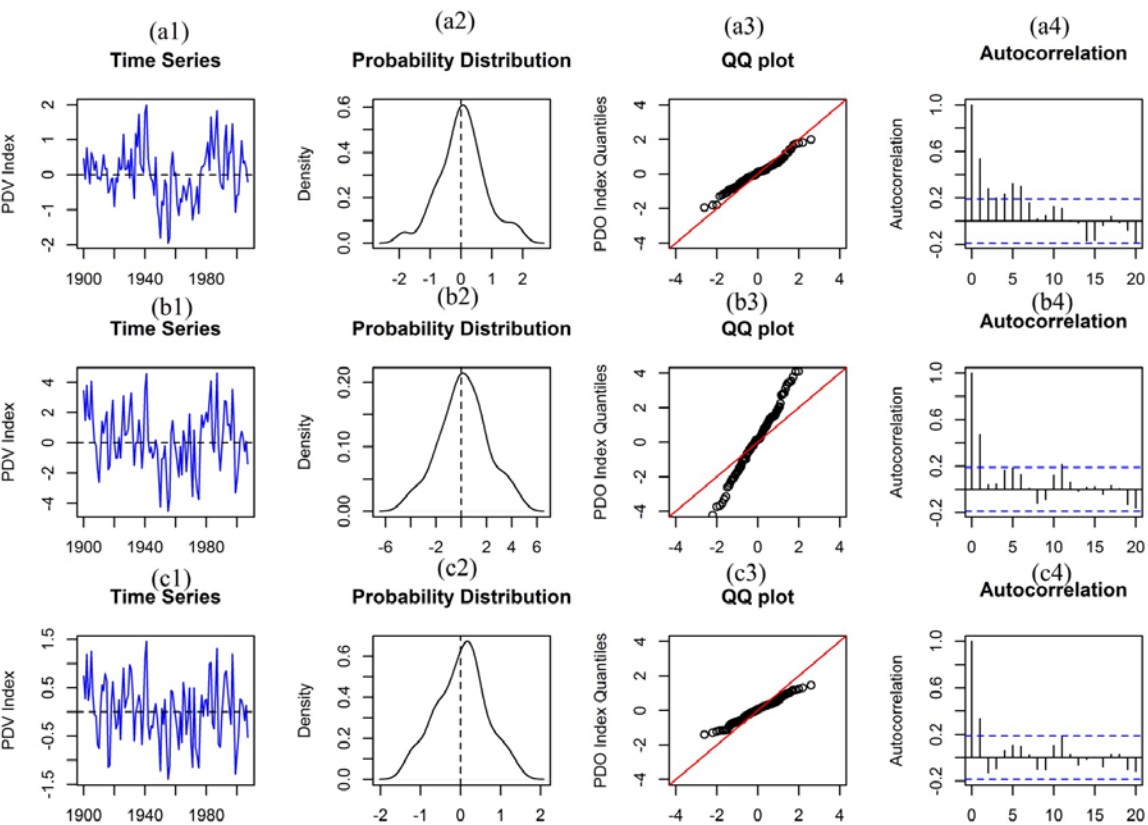

Figure 1 Annual instrumental time series, probability density plot, QQ plot and autocorrelation plot: (a): PDO from Mantua (1900-2015); (b): IPO from Parker (1871-2007); (c): IPO from Henley (1870-2007); The dashed lines in (a4)-(c4) present the 95% confidence bands for zero autocorrelation.

2    **Figure 2 Run length time series extracted using the dynamic and static threshold methods for**

3    **reconstructions with centennial trends**

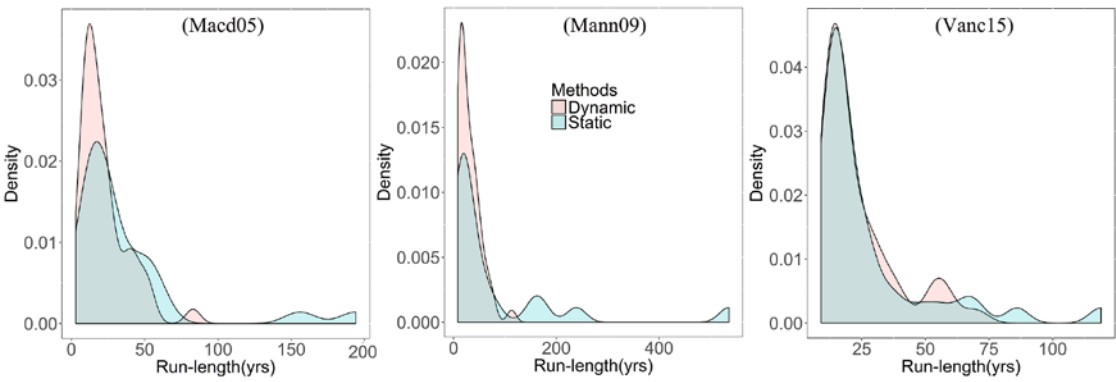

2    **Figure 3 Comparison of run length distributions extracted using the dynamic and static threshold**

3    **methods in reconstructions with centennial trends**

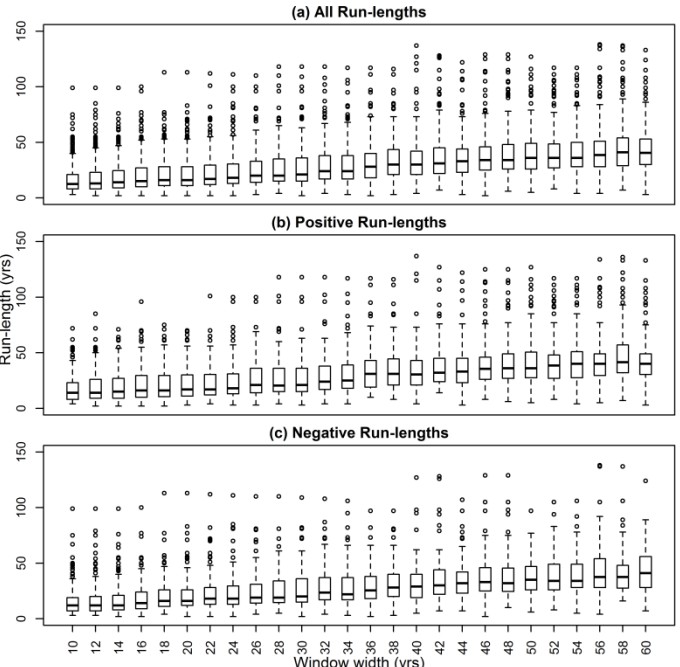

3 **Figure 4 Boxplot of run lengths samples from all reconstructions with different window width in Mann-**

4 **Whitney method**

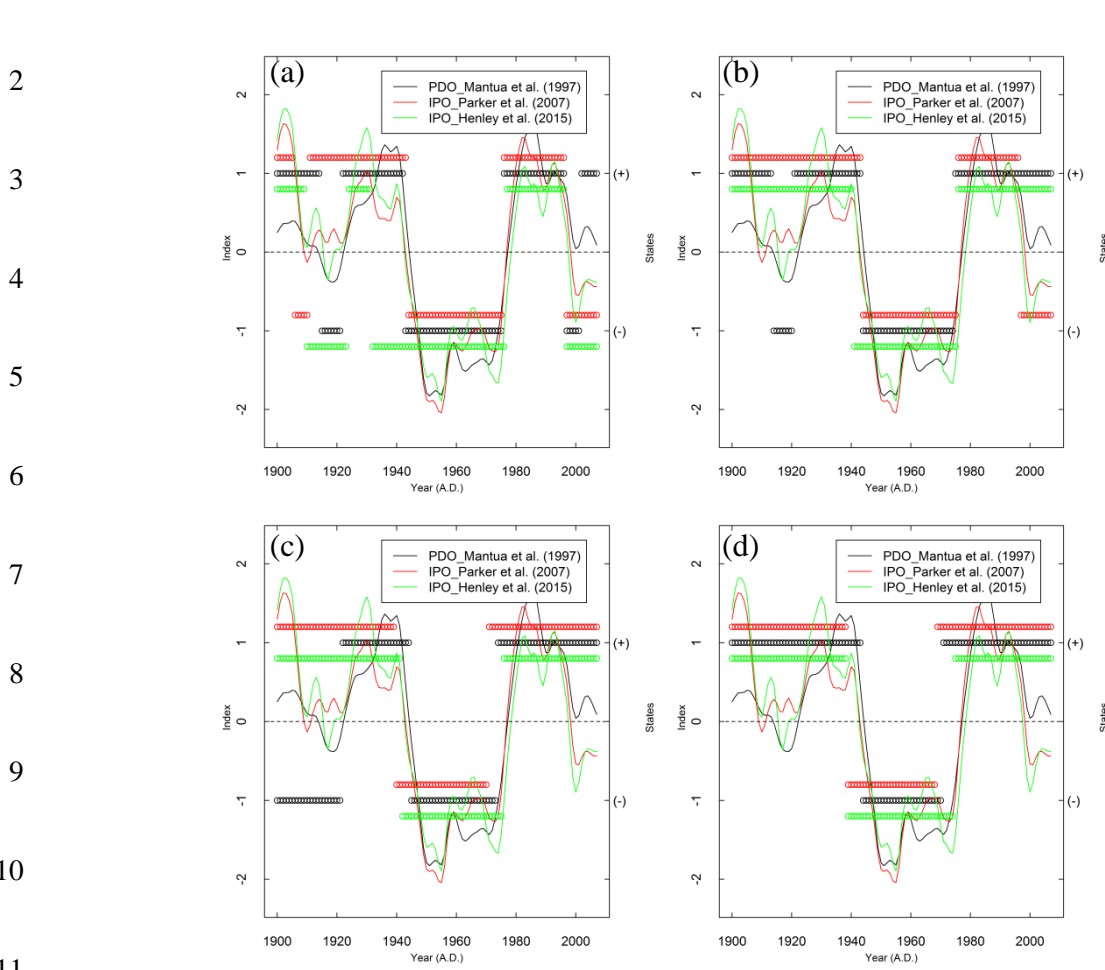

12    **Figure 5 Time series plots of different instrumental PDV indices and corresponding run lengths**

13    **(represented by dots in corresponding colour) extracted using different window: (a) 10; (b) 20; (c) 30; (d)**

14    **40 years**

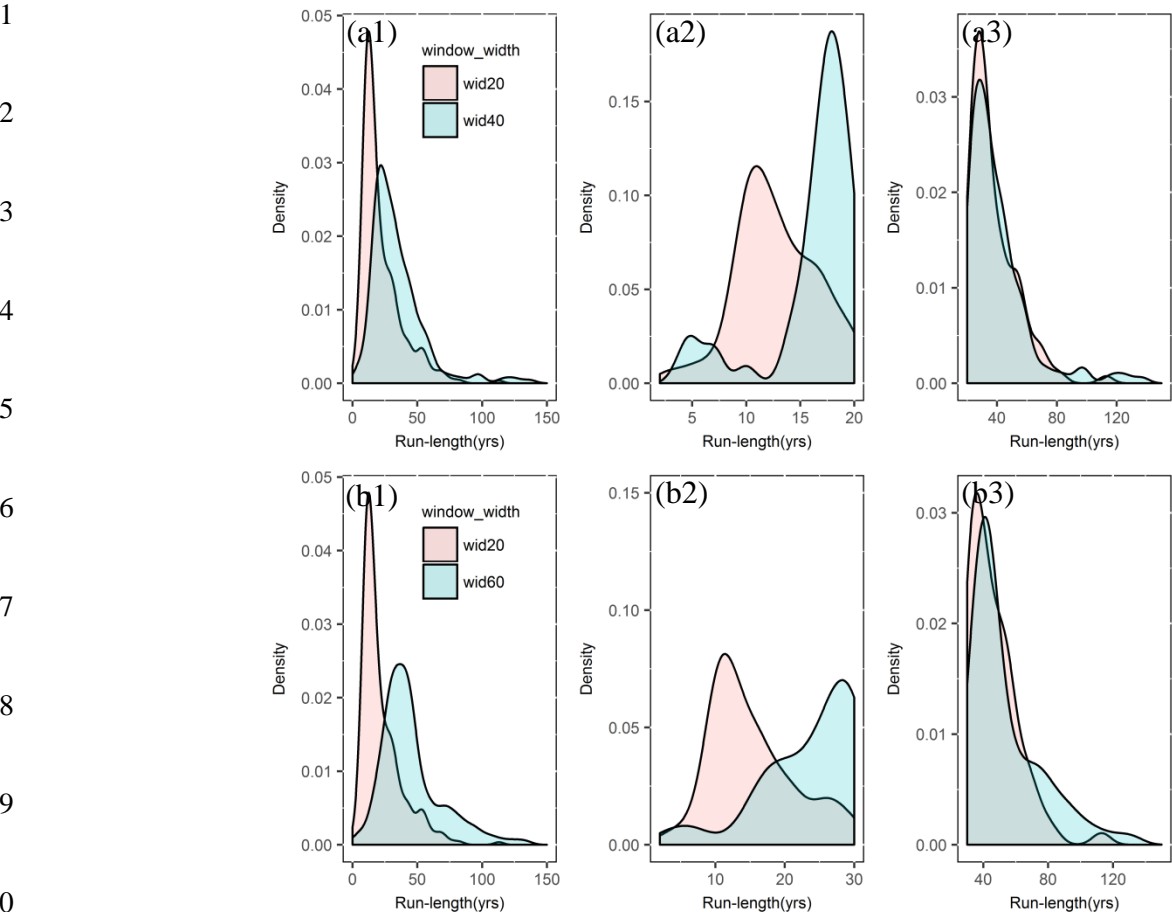

**Figure 6 Density plot of run length samples by applying different window widths to all reconstructions, (a1-a3): comparison between window width 20 and 40, in which (a1) is unconditional run length distribution, panel (a2) is the distribution conditioned on run lengths less than 20 years and panel (a3) is the distribution conditioned on run lengths greater than 20 years; (b1-b3): comparison between window width 20 and 60, in which (b1) is unconditional run length distribution, panel (b2) is the distribution conditioned on run lengths less than 30 years and panel (b3) is the distribution conditioned on run lengths greater than 30 years**

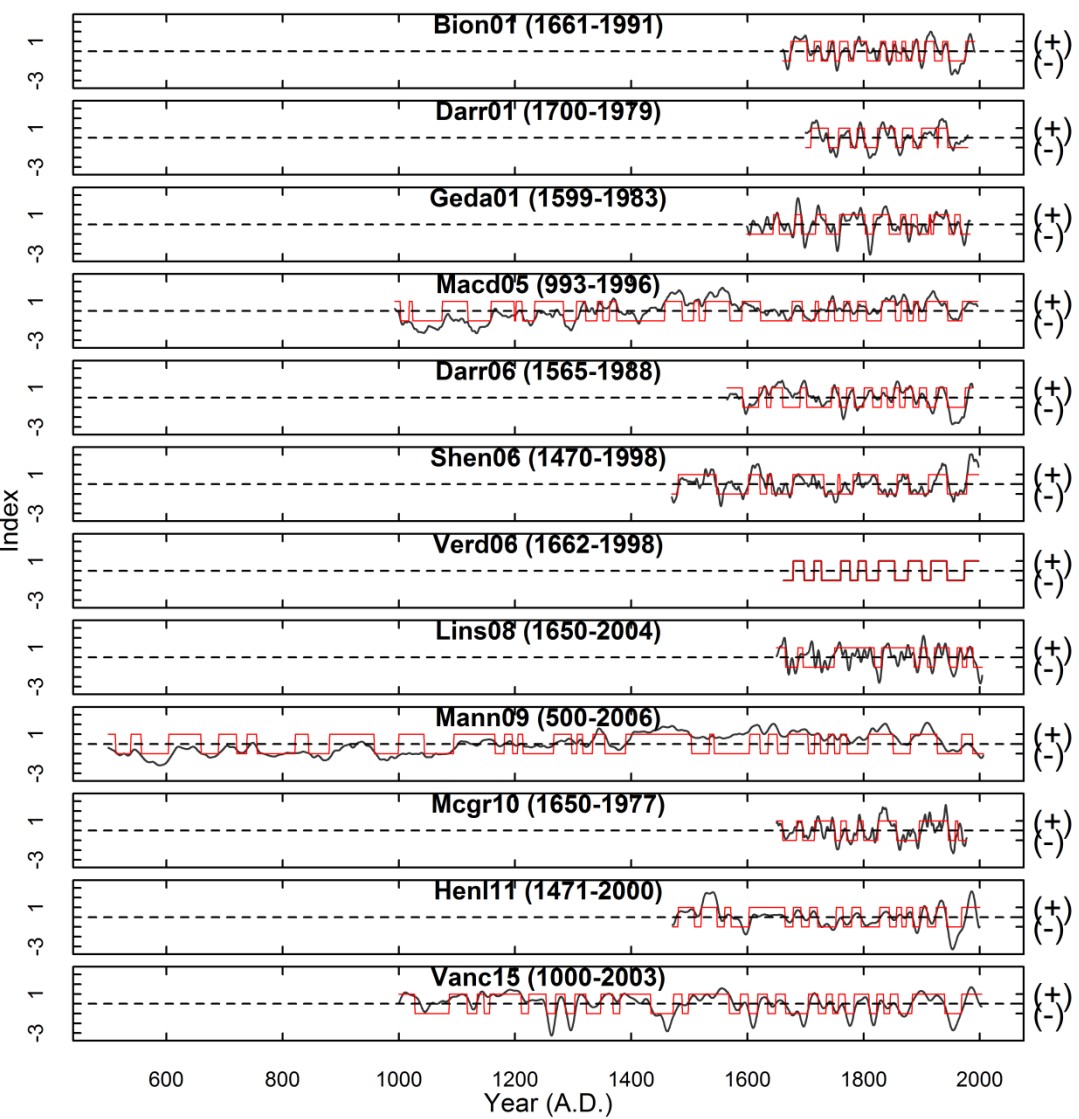

2 **Figure 7 Filtered PDV reconstruction time series (black line) with extracted run length (red line)**

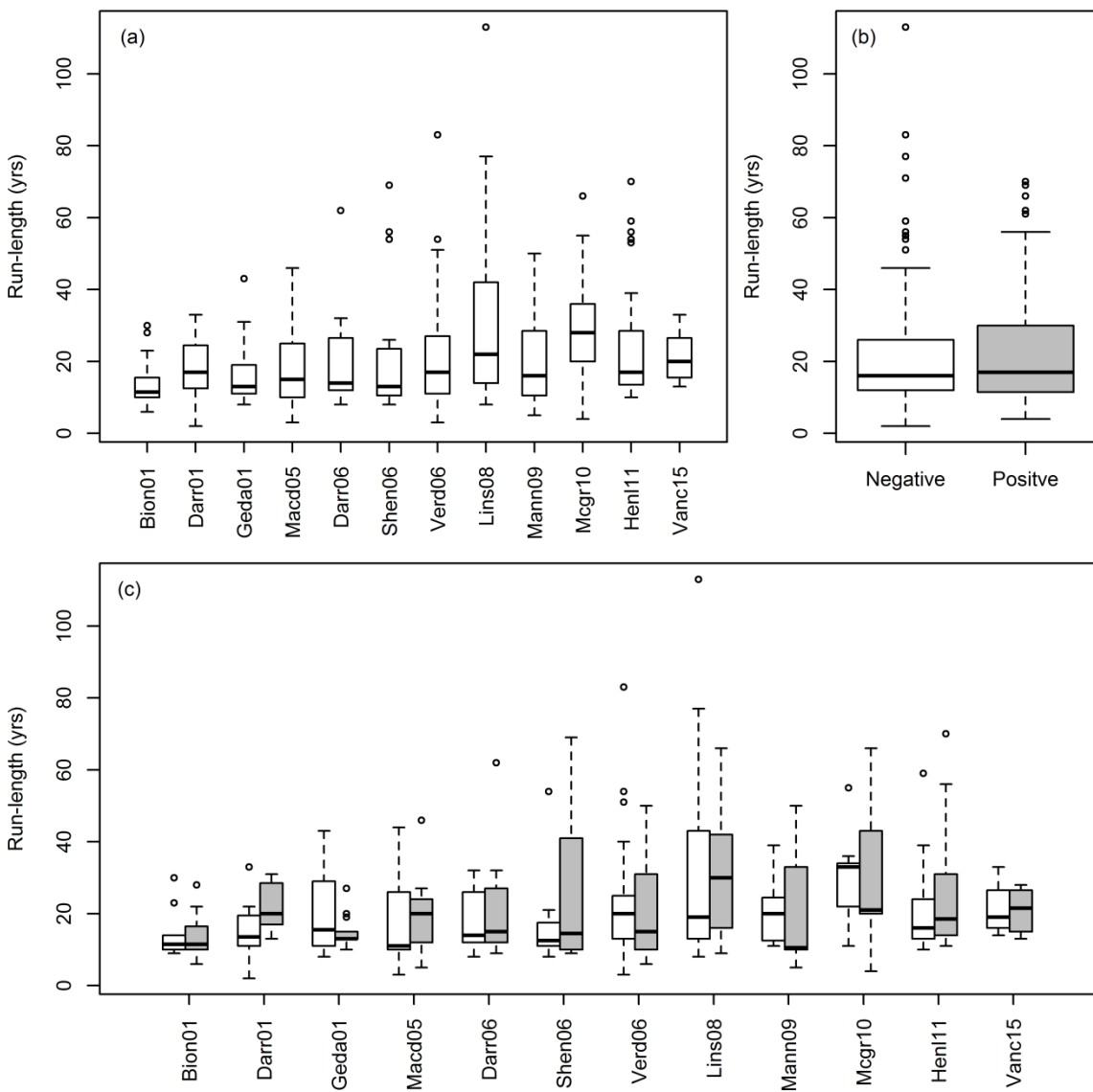

**Figure 8 Boxplot of run lengths samples from different reconstructions: (a) all run lengths for each reconstruction; (b) pooled positive (grey) and negative (white) run lengths; (c) positive (grey) and negative (white) run lengths for each reconstruction**

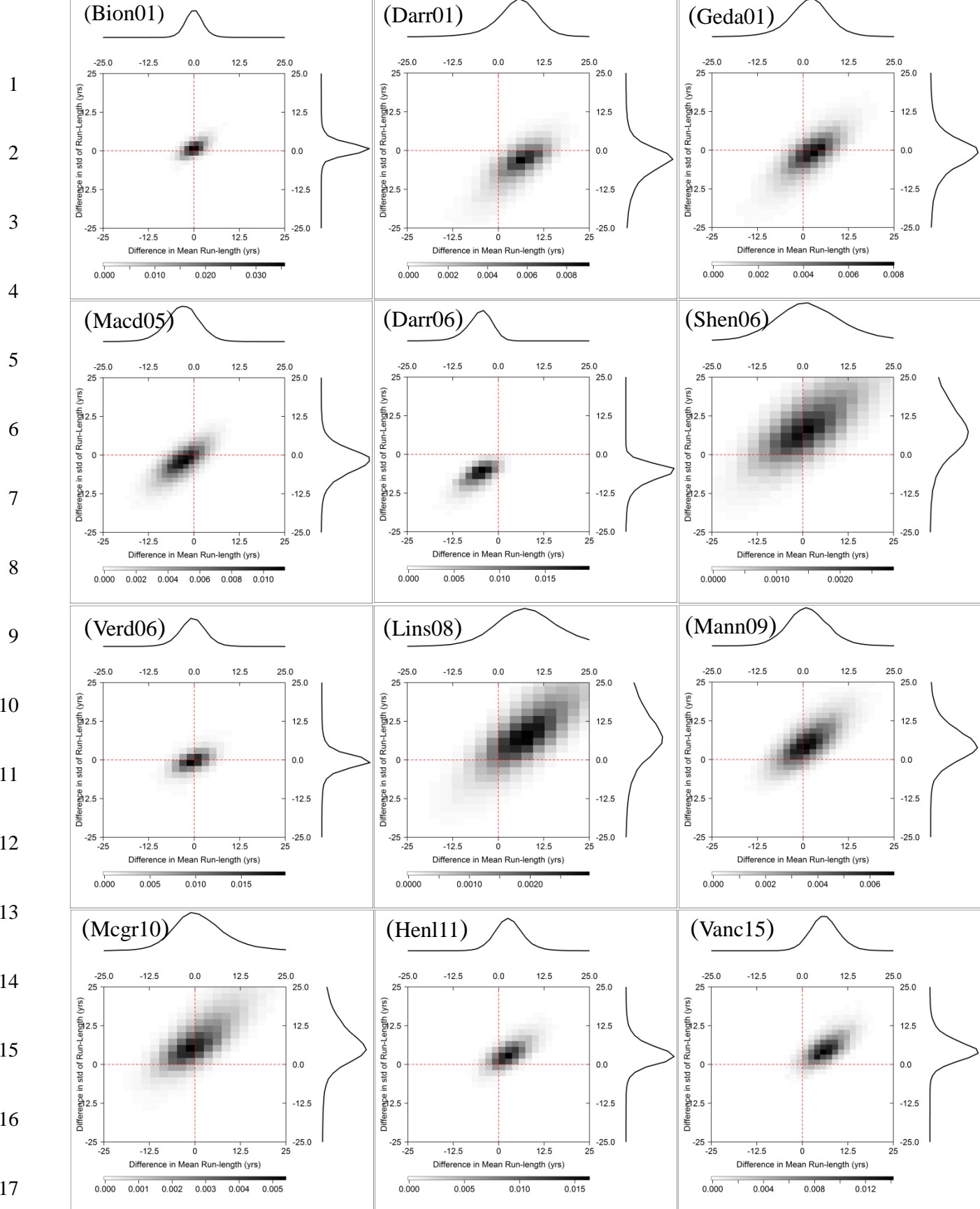

**Figure 9 Density and scatter plots of the differences between mean and standard deviation of positive and negative run length simulations (positive minus negative)**

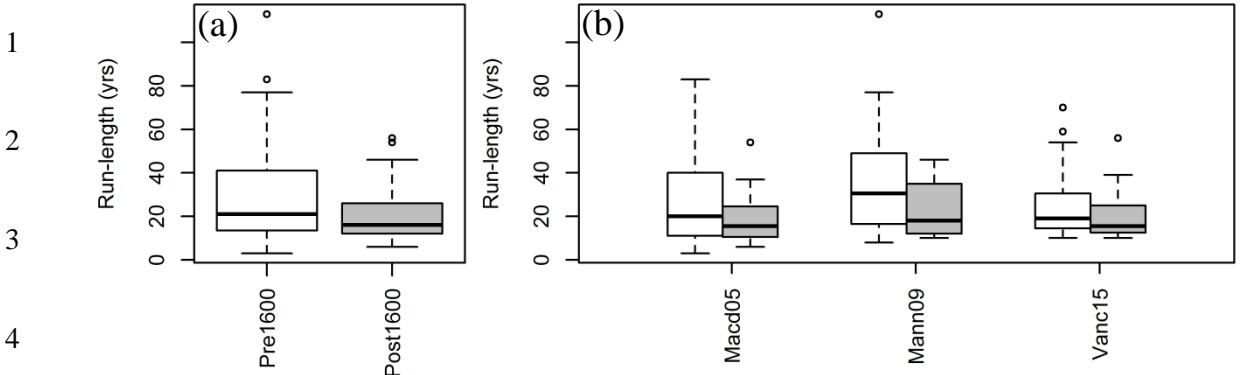

**Figure 10 Boxplot of run lengths samples from different reconstructions (white box indicates pre-1600 and grey box indicates post-1600): (a) pooled run lengths from all three reconstructions; (b) run lengths for each reconstruction**

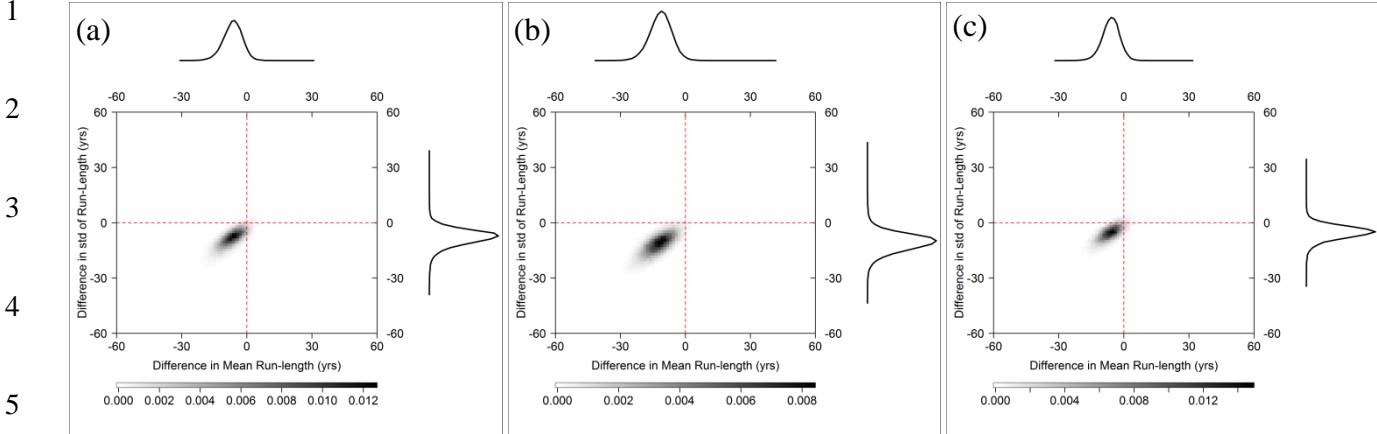

**Figure 11 Density and scatter plots of the differences between mean and standard deviation of pre-1600 and post-1600 run length simulations (post-1600 minus pre-1600): (a) Macd05; (b) Mann09; (c) Vanc15**