# Peer review of "Using paleoclimate reconstructions to analyse"

_Hydrology and Earth System Sciences, 2018_

## Referee Comment (RC1) · Anonymous Referee #1 · 18 May 2018

This paper provides very interesting results, and topic of the research is within the scope of this journal. The manuscript is well written and organized, but I have several minor comments as follows.

[General Comments] In the dynamic threshold method, three parameters (windows width: w, cut-off frequency: y and confidence level: alpha) should be specified, and sensitivity of w is discussed in this study. However, how about the sensitivity of y and alpha on extracting run length? # If authors could provide some comments/ideas on that, it would be helpful for readers. # (but do not have to conduct additional sensitivity analysis.)

[Figure]

[Specific comments]

P13, P24-25: How did you set the value of "confidence level alpha (=90%)" and "cutting-off frequency (=1/11)" for this analysis? A little more explanation is expected (if possible). # P10, L14: y=100 (?)

P15, L13- "The absence of a consistent difference ... may be a consequence of sampling variability ... " –> The meaning of this part is a little unclear. # What is "sampling variability"?

P28, Fig.1 What blue broken lines in (a4), (b4) and (c4) of Fig.1 are representing? # 95% confidence bands for zero autocorrelation (as described P7, L8)?

---

## Referee Comment (RC2) · Anonymous Referee #2 · 25 Jun 2018

General comments

This is an interesting and mostly well-written manuscript with a topic of interest to HESS. My main comments concerns the presentation, which I think can be strengthened in places; in particular as the authors propose a new methods.

On page 11, lines 20-23 it is stated that Vanc15 is more reliable than Macd05 and Mann09. As such, does the proposed threshold method help to cover-up problems with the relatively poor quality of some of the reconstructions? If one of the reconstructions is clearly more reliable than others, should the advice not be to use Vanc15 in future studies and not Macd05 or Mann09?

[Figure]

Is there a case for also looking at the magnitude of sojourns above and below a threshold in addition to duration?

Other comments

Page 2, line 14-15: semantics, but do you mean the impact of PDV has been explored at locations around the world?

Page 3, lines 8-10: Would be useful to see geographical extend and topic (flood/drought) for each of these references.

Page 4, line 11: what is the unit for the +-0.5 threshold? See also page 8, line 12.

Page 6, line 21: SST not defined.

Page 7, line 21: define paleo proxies.

Page 8, lines 8-15: This paragraph is a very similar to page 4. Is it necessary to write all this twice?

Page 8, lines 22-23: As per my comment in the introduction: is the use of a more dynamic threshold method simply masking underlying problems with some of the reconstructed datasets. If so, should these datasets not be excluded from further analysis on the basis that more reliable datasets are now available?

Page 9: I am not sure I understand the method to a level where I could implement my own version of the dynamic threshold framework; in particular, step 1. Maybe a conceptual figure assisting the reader could be useful here?

Page 9: I have not previously come across a Butterworth filter. Are Eqs 1 and 2 filters of this type?

Page 10, line 4: unit on y=100? Section 3.2.2: Last sentence (lines 11-13) is rather meaningless and not necessary I think.

Page 13: I am not clear on what is the difference between the conditional and unconditional distributions.

Figure 6: There is a lot of information on this figure but the overall summary is summarised nicely on page 17-19. Could this figure somehow be simplified to be more focussed on this message?

Page 13, line 24: Is alpha not a significance level (\alpha=10%) rather than a confidence level as it refers back to a statistical test?

Page 13, line 25: On page 10, line 14 y is defined as 100, but here y=11; why the difference?

Section 4: Not sure the title of this section is appropriate as 'hydrology' is not discussed at any point.

Page 15, line 9: What does 'pooled' mean? Is this run-lengths extracted from all 12 reconstructions merged together into one sample?

Page 15-16: There is a lot of information on Figure 9, but I am not sure I understand how to interpret them. Also, are the conclusions derived from Figure 9 really that different from what you have already found in the much more easily interpreted Figure 8, that there is little evidence of statistical significant differences? Same comment on the difference between Figures 10 and 11.

Page 15: Symbols \mu+, \mu-, \sigma+ and \sigma- are not defined anywhere?

Page 18, lines 4-5. Not sure I understand the meaning of this sentence.

---

## Author Comment (AC1) · 25 Jun 2018

Comments: This paper provides very interesting results, and topic of the research is within the scope of this journal. The manuscript is well written and organized, but I have several minor comments as follows. [General Comments] In the dynamic threshold method, three parameters (windows width: w, cut-off frequency: y and confidence level: alpha) should be specified, and sensitivity of w is discussed in this study. However, how about the sensitivity of y and alpha on extracting run length? # If authors could provide some comments/ideas on that, it would be helpful for readers. # (but do not have to conduct additional sensitivity analysis.)

[Figure]

Reply: We thank the reviewer for the comments and appreciate the opportunity to clarify aspects of the manuscript. Below we present our responses and indicate the changes made to the original manuscript.

The cut-off frequency parameter in the dynamic threshold method is used to filter out centennial trends that may be mixed with decadal variability. We therefore believe it is appropriate to use a cut-off frequency of 1/100 years. This parameter is aimed at filtering centennial trends and has little influence on the inferred runs which have significantly higher frequencies.

We have replaced Line 12 Page 10 in the original manuscript with "The cut-off frequency parameter in the dynamic threshold method is used to filter out centennial trends that may be mixed with decadal variability and should have little influence on the statistical characteristics of the inferred runs. A cut-off frequency of 1/100 years is considered to adequately meet these requirements."

The Mann-Whitney test is used to determine whether two independent sets of data come from the same distribution. There is a (100-$\alpha$)% percent chance of the test statistic falling outside the $\alpha$% confidence limits under the null hypothesis. If $\alpha$ is selected close to 100 and the test statistic falls outside the confidence limits, we are confident that the detected change point is actually a change point. The value of $\alpha$ is typically set by the researcher and involves consideration of type 1 and 2 errors (Zar, 1999). In this study, we follow the practice used in other reconstruction studies (Gedalof and Smith, 2001; Shen et al., 2006; McGregor et al., 2010; Pent et al., 2015) that adopt a confidence level of 90% so that the chance of making a type 1 error (that is, rejecting a changepoint when in fact there is one) is low, namely 10%.

We have added the following lines in the original manuscript at Line 14 Page 10 "In this study, we set the confidence level to be 90%, a value that is consistent with other reconstruction studies (eg Gedalof and Smith, 2001; Shen et al., 2006; McGregor et al., 2010; Pent et al., 2015)".

[Specific comments] P13, P24-25: How did you set the value of "confidence level alpha (=90%)" and "cutting-off frequency (=1/11)" for this analysis? A little more explanation is expected (if possible). # P10, L14: y=100 (?)

Reply: See the above reply for our response on setting of the confidence level $\alpha$.

The cut-off frequency of 1/11 is only used in Figure 7 to plot filtered PDV reconstructions. It is not used in the dynamic threshold method. The cut-off frequency of 1/11 was selected to be consistent with the cut-off frequency used in the Parker instrumental IPO time series.

P15, L13- "The absence of a consistent difference ... may be a consequence of sampling variability ... " –> The meaning of this part is a little unclear. # What is "sampling variability"?

Reply: Random samples from a given population are used to estimate target statistics for this population. Sampling variability refers to the variability in the target statistics that arises when random sampling is repeated. For example, two samples of 10 runs from the true run population will give different estimates of the run standard deviation. In this research, there are a limited number of PDV run samples. Therefore, it is important to recognize that differences in run statistics may be due to sampling variability. It is only when differences are bigger than what would be expected due to sampling variability that we would conclude there is a significant difference.

We have added the following text at Line 15 Page 15 to clarify this important issue:" Sampling variability refers to the variability in the target statistics that arises when random sampling is repeated. Therefore, it is important to recognize that differences in run statistics may be due to sampling variability. It is only when differences are bigger than what would be expected due to sampling variability that we would conclude there is a significant difference."

P28, Fig.1 What blue broken lines in (a4), (b4) and (c4) of Fig.1 are representing? #

95% confidence bands for zero autocorrelation (as described P7, L8)?

Reply: The dashed lines present the 95% confidence bands for zero autocorrelation. If the true autocorrelation were zero, there is 95% chance that the sample autocorrelation coefficient will lie between the dashed lines.

We have added the following text in Line 11 Page 28 Figure 1 caption: " The dashed lines in (a4)-(c4) present the 95% confidence bands for zero autocorrelation."

Reference:

Gedalof, Z., and Smith, D. J.: Interdecadal climate variability and regime-scale shifts in Pacific North 32 America, Geophysical Research Letters, 28, 1515-1518, 10.1029/2000GL011779, 2001. McGregor, S., Timmermann, A., and Timm, O.: A unified proxy for ENSO and PDO variability since 20 1650, Climate of the Past, 6, 1-17, 10.5194/cp-6-1-2010, 2010. Peng, Y., Shen, C., Cheng, H., and Xu, Y.: (2015): Simulation of the Interdecadal Pacific Oscillation and its impacts on the climate over eastern China during the last millennium. J. Geophys. Res. Atmos., 120, 7573–7585. doi: 10.1002/2015JD023104. Shen, C., Wang, W. C., Gong, W., and Hao, Z.: A Pacific Decadal Oscillation record since 1470 AD 46 reconstructed from proxy data of summer rainfall over eastern China, Geophysical Research Letters, 47 33, L03702, 10.1029/2005GL024804, 2006. Zar, J. H. (1999). Biostatistical analysis, 663 pp: Prentice Hall, Englewood Cliffs.

Please also note the supplement to this comment:
https://www.hydrol-earth-syst-sci-discuss.net/hess-2018-173/hess-2018-173-AC1-supplement.pdf

---

## Author Comment (AC2) · 29 Aug 2018

General comments This is an interesting and mostly well-written manuscript with a topic of interest to HESS. My main comments concerns the presentation, which I think can be strengthened in places; in particular as the authors propose a new methods. On page 11, lines 20-23 it is stated that Vanc15 is more reliable than Macd05 and Mann09. As such, does the proposed threshold method help to cover-up problems with the relatively poor quality of some of the reconstructions? If one of the reconstructions is clearly more reliable than others, should the advice not be to use Vanc15 in future studies and not Macd05 or Mann09?

[Figure]

Reply: We thank the reviewer for the comments which have helped improve the manuscript and appreciate the opportunity to clarify aspects of the manuscript. Below we present our responses and indicate the changes made to the original manuscript.

The threshold method does not hide problems with poor quality reconstructions. Indeed Figure 7 clearly shows the reconstructions produce quite different run time series. However, reconciling these reconstructions was not the objective of the paper. Rather the objective here is to develop a robust run length extraction method to maximize the potential value of the reconstructions and to investigate the consistency of the statistical signatures with regard to two hydrologically important questions,: (i) are PDV runs lengths stationary and (ii) is the persistence (i.e. run lengths) of wet and dry epochs different.

In the last paragraph of the conclusions, we comment that the reconstructions provide quite different PDV run length time series and that reconstructing the palaeo PDV run length time series from multiple reconstructions will need careful consideration of errors. For example, the approach presented by Henley et al. (2011) combines individual reconstructions according to their accuracy. However, the accuracies are not high with Nash-Sutcliffe indeces < 0.48, suggesting even favouring the more accurate individual reconstructions still results in a combined/overall reconstruction with large errors. So our approach is to look for a signal that is consistent in all the individual reconstructions.

We have added the following text after Line 23 Page 11 in the original manuscript: "Individual reconstructions are subject to different error structures and magnitudes. Use of multiple reconstructions can help reduce the impact of errors, and thus provide more insight into the statistical characteristics of PDV run lengths. Nonetheless, errors still need to be carefully considered. This is illustrated by Henley et al. (2011) who combines individual reconstructions according to their accuracy. However, the accuracies of the individual reconstructions are not high (Nash-Sutcliffe indices <0.48) suggesting that even favouring the more accurate individual reconstructions will still result in a combined/overall reconstruction with large errors. Therefore, the focus of our research

is to identify a signal (or signals) that is (are) common to all individual reconstructions."

On Line 18 Page 4 we cited von Storch et al. (2004, 2006) and Wahl et al. (2006) who considered the problem of whether to use detrended or nondetrended data. We have added the following sentences on Line 18 Page 4 to expand the discussion of von Storch et al. (2004, 2006) and Wahl et al. (2006). "As stated by von Storch et al. (2006), "it is commonly accepted that proxy indicators may contain nonclimatic trends". Moreover, the calibration and validation of any statistical method using nondetrended data may be compromised because the nonclimatic trends may be interpreted as a climate signal. The centennial trends in PDV reconstructions may be either nonclimatic trends or non-decadal climate trends. Whichever the case, it is necessary to filter such centennial trends before interpreting decadal climate variability."

Is there a case for also looking at the magnitude of sojourns above and below a threshold in addition to duration?

Reply: We agree that looking at the magnitude of sojourns above/below a threshold would be of considerable interest. However, that is not the focus here as we are aiming to utilize PDV run length information to parameterise stochastic models used in water resources management related decision making.

Other comments Page 2, line 14-15: semantics, but do you mean the impact of PDV has been explored at locations around the world?

Reply: Yes the impact of PDV has been explored at various locations around the world. As the impact of PDV is discussed in the paragraph starting Line 3 Page 3, we have deleted "and explored" in Line 13 Page 2.

Page 3, lines 8-10: Would be useful to see geographical extend and topic (flood/drought) for each of these references.

Reply: As the focus of the paper is not the PDV impact, which is now established, we feel the paragraphs starting at Line 12 Page 2 and Line 3 Page 3 provide sufficient

information about the geographic spread of PDV and its impact context.

Page 4, line 11: what is the unit for the +-0.5 threshold? See also page 8, line 12.

Reply: The threshold is unitless as the PDV indices are standardized. To make this explicit, in Line 11 Page 4 "Vance et al. (2015) used $\pm 0.5$ as thresholds" is replaced with "After standardizing the indices, Vance et al. (2015) used $\pm 0.5$ as thresholds"

Page 6, line 21: SST not defined.

Reply: Agreed. We replaced Line 21 Page 6 "monthly SST anomalies" with "monthly Sea Surface Temperature (SST) anomalies"

Page 7, line 21: define paleo proxies.

Reply: The original text in Line 21 Page 7 "The reconstructions are based on paleo proxies" is replaced with "The reconstructions are based on paleo proxies (i.e. preserved physical characteristics of the environment that can be directly measured)".

Page 8, lines 8-15: This paragraph is a very similar to page 4. Is it necessary to write all this twice?

Reply: Agreed.

Changes to original manuscript: Page 8, Lines 8-15 are deleted and Page 8 Line 16-17 are changed to "For reconstructions Macd05, Mann09 and Vanc15 which exhibit centennial trends (see Table 1 and Figure 2), the above mentioned static threshold methods used in Verdon and Franks (2006), Henley et al. (2011), Vance et al. (2015) and Henley et al. (2017) may not identify meaningful decadal phases."

Page 8, lines 22-23: As per my comment in the introduction: is the use of a more dynamic threshold method simply masking underlying problems with some of the reconstructed datasets. If so, should these datasets not be excluded from further analysis on the basis that more reliable datasets are now available?

Reply: The dynamic threshold method only filters out centennial trend and has little influence on the inferred run lengths which have significantly higher frequencies. Please refer to the reply to the first comment for further commentary on this.

Page 9: I am not sure I understand the method to a level where I could implement my own version of the dynamic threshold framework; in particular, step 1. Maybe a conceptual figure assisting the reader could be useful here?

Reply: We explored a number of options but felt our efforts would not value add beyond the existing four-step description and Figure 2 which shows the PDV time series and phases determined using the dynamic threshold method.

Instead we have opted to provide the R code for the dynamic threshold method in supplementary information. Furthermore we have enhanced the description of step 1 as follows: "1. Detect step-change points. For a given reconstruction, apply a change point detection method. A number of methods can be used. In this study we used the non-parametric Mann-Whitney test method (Mauget, 2003) with a given window width w and confidence level $\alpha$ to identify the step-change points. The method involves centring a window of width w at a particular year t and then applying the Mann-Whitney test to the samples of width w/2 at or before and after year t. A step change is deemed to occur if the Mann-Whitney test statistic lies outside the $\alpha$ confidence limits (under the null hypothesis)."

In addition, we have referred to Figure 2 earlier. In Line 3 Page 9 we have added: "Figure 2 provides insight about the mechanics of dynamic threshold method showing the PDV time series and the resulting block phase waveform."

Page 9: I have not previously come across a Butterworth filter. Are Eqs 1 and 2 filters of this type?

Reply: Eq 1 is the arithmetic mean of the PDV index for a particular run. It does not use the Butterworth filter. Likewise Eq 2 is an arithmetic mean. However, in this

case, it is taking the average of the Butterworth filtered index values. We chose the Butterworth filter to maintain consistency with Henley et al. (2011) who selected the filter on account of its near-flat frequency response in the pass band. In Line 15 Page 9 we replace "(Selesnick and Burrus, 1998)" with "(which was used by Henley et al. (2011) to filter paleo PDV indices)".

Page 10, line 4: unit on y=100? Section 3.2.2: Last sentence (lines 11-13) is rather meaningless and not necessary I think.

Reply: In Page 9 Line 17 it is stated that "cut-off frequency 1/y year-1", in which y is a unitless parameter. Changes to original manuscript: Last sentence (lines 11-13) in Section 3.2.2 is deleted.

Page 13: I am not clear on what is the difference between the conditional and unconditional distributions.

Reply: We have removed reference to "unconditional" but have retained usage of conditional. A conditional distribution is the distribution of runs that meet a specific condition. For example, a distribution conditioned on runs less than 20 years would exclude all runs greater than 20 years.

Changes to original manuscript: Page 13 Line 8-11 are changed to "Panel (a1) shows the empirical density of run lengths, while panels (a2) and (a3) show the empirical densities conditioned on run lengths less than 20 years and greater than 20 years respectively. Likewise Panels (b1) to (b3) repeat this sequence for 20 and 60-year windows with the conditioning threshold of 30 years".

Figure 6: There is a lot of information on this figure but the overall summary is summarised nicely on page 17-19. Could this figure somehow be simplified to be more focussed on this message?

Reply: We thank the referee for the positive comment on our summary. However, we cannot see any way to further simplify Figure 6 - removing any feature from the plot

would leave some of our conclusions unsubstantiated. The plots highlight the critical influence of window width on the inferred run length distributions.

Page 13, line 24: Is alpha not a significance level (\alpha=10%) rather than a confidence level as it refers back to a statistical test?

Reply: We accept that our usage of "level" may cause confusion. We have replaced "confidence level" with "confidence limits". As explained in an earlier response, if the test statistic falls outside its $\alpha$ confidence limits (under the null hypothesis), a step change is deemed to have occurred.

Page 13, line 25: On page 10, line 14 y is defined as 100, but here y=11; why the difference?

Reply: The cut-off frequency of 1/11 is only used in Figure 7 to plot filtered PDV reconstructions. It is different to the parameter y in the dynamic threshold method, which is used for filter out centennial trend. The cut-off frequency of 1/11 was selected to be consistent with the cut-off frequency used in the Parker instrumental IPO time series.

Section 4: Not sure the title of this section is appropriate as 'hydrology' is not discussed at any point.

Reply: In the introduction we stated that positive/negative PDV phases are related to dry/wet hydrological epochs. This association motivated the original title. Nonetheless, we accept the reviewer's point and have changed the section title to "Statistical signatures of PDV run lengths".

Page 15, line 9: What does 'pooled' mean? Is this run-lengths extracted from all 12 reconstructions merged together into one sample?

Reply: Yes, pooling does mean that all 12 reconstructions are merged into a single sample set. Please note pooled positive runs refer to merging all the positive runs in the 12 reconstructions into a single sample set.

Page 15-16: There is a lot of information on Figure 9, but I am not sure I understand how to interpret them. Also, are the conclusions derived from Figure 9 really that different from what you have already found in the much more easily interpreted Figure 8, that there is little evidence of statistical significant differences? Same comment on the difference between Figures 10 and 11.

Reply: Sampling variability makes interpreting differences particularly challenging when working with small samples, the danger is to avoid attributing significance to what is likely to be noise. For this reason, we feel that analysis of differences from different perspectives helps guard against such an event. The fact that the conclusions drawn from Figure 9 are in line with Figure 8 (same with Figures 10 and 11) provides more confidence. For that reason, we believe Figures 9 and 11 offer value and should be retained.

Page 15: Symbols mu+, mu-, sigma+ and sigma- are not defined anywhere?

Reply: We changed Page 15 Line 22 into "Figure 9 presents the joint posterior distribution of the differences in mean ($\mu$+-$\mu$−) and standard deviation ($\sigma$+−$\sigma$−) for each reconstruction"

Page 18, lines 4-5. Not sure I understand the meaning of this sentence.

Reply: We have replaced the first sentence is section 4.3 with "All of the reconstructions reported in Table 1 did not distinguish between positive and negative PDV runs in their analysis, except for Vance et al. (2015)."